# fMRI Evidence for Default Mode Network Deactivation Associated with Rapid Eye Movements in Sleep

**DOI:** 10.3390/brainsci11111528

**Published:** 2021-11-18

**Authors:** Charles Chong-Hwa Hong, James H. Fallon, Karl J. Friston

**Affiliations:** 1Patuxent Institution, Correctional Mental Health Center—Jessup, Jessup, MD 20794, USA; 2Department of Psychiatry and Behavioral Sciences, The Johns Hopkins University, Baltimore, MD 21205, USA; 3Department of Anatomy and Neurobiology, University of California, Irvine, CA 92697, USA; jfallon@uci.edu; 4Department of Psychiatry and Human Behavior, University of California, Irvine, CA 92697, USA; 5The Well Come Centre for Human Neuroimaging, Institute of Neurology, University College London, London WC1N 3AR, UK; k.friston@ucl.ac.uk

**Keywords:** default mode network (DMN), rapid eye movements (REMs) in sleep, hierarchical predictive coding, visual perception, dream, retrosplenial cortex, autism, functional MRI (fMRI), hallucinogen

## Abstract

System-specific brain responses—time-locked to rapid eye movements (REMs) in sleep—are characteristically widespread, with robust and clear activation in the primary visual cortex and other structures involved in multisensory integration. This pattern suggests that REMs underwrite hierarchical processing of visual information in a time-locked manner, where REMs index the generation and scanning of virtual-world models, through multisensory integration in dreaming—as in awake states. Default mode network (DMN) activity increases during rest and reduces during various tasks including visual perception. The implicit anticorrelation between the DMN and task-positive network (TPN)—that persists in REM sleep—prompted us to focus on DMN responses to temporally-precise REM events. We timed REMs during sleep from the video recordings and quantified the neural correlates of REMs—using functional MRI (fMRI)—in 24 independent studies of 11 healthy participants. A reanalysis of these data revealed that the cortical areas exempt from widespread REM-locked brain activation were restricted to the DMN. Furthermore, our analysis revealed a modest temporally-precise REM-locked decrease—phasic deactivation—in key DMN nodes, in a subset of independent studies. These results are consistent with hierarchical predictive coding; namely, permissive deactivation of DMN at the top of the hierarchy (leading to the widespread cortical activation at lower levels; especially the primary visual cortex). Additional findings indicate REM-locked cerebral vasodilation and suggest putative mechanisms for dream forgetting.

## 1. Introduction

The default mode network (DMN) is a highly correlated network of brain regions comprising medial prefrontal cortex, lateral superior and inferior frontal gyrus, rostral anterior cingulate, posterior cingulate cortex (PCC), precuneus (PCu), retrosplenial cortex (RSC), posterior inferior parietal cortex (IPC), angular gyrus, temporoparietal junction, temporal pole, hippocampus, parahippocampus, and lateral temporal cortex [1,2,3]. This intrinsic brain network activates by default during rest and when attention to sensory input is precluded [4]. Functional connectivity among the brain regions of the DMN—that was originally described during the awake resting state [5,6] and sensory visual processing [5]—persists during light sedation [7], deep anesthesia [8], light [9,10] and deep [11] non-rapid eye movement sleep. It is likely that the intrinsic organization and reciprocal fluctuations in the task-positive network (TPN) and DMN [6] also persist in REM sleep. Indeed, this anticorrelation has been confirmed [12]. This prompted us to reanalyze our event-related (event being REMs) functional MRI (fMRI) data collected with synchronous timing (from video recording) of REMs in sleep [13] to characterize DMN-TPN anticorrelations in relation to temporally-precise REM events, rather than epochs of REM sleep. 

REMs are the hallmark of REM sleep, but a substantial portion of REM sleep does not contain REMs. We have previously established [13] that *brain activation time-locked to REMs is characteristically widespread* (at *p* < 0.05, corrected for multiple comparisons)—compared to the clearly circumscribed activation time-locked to finger tapping, auditorily cued and timed after occurrence of REMs at the same threshold. Such widespread activation is unusual in mass univariate analyses of this sort [13], and speaks to REM-locked distributed, global processing [14,15]. Crucially, within the widespread activation, robust (corrected *p* < 0.0005) *peak responses are seen in the primary and secondary visual cortices and the multisensory binding system* [13]. These findings speak to REM-locked hierarchical processing of visual information, where each sensory modality is processed in its respective primary sensory cortex, with ensuing multisensory integration at higher hierarchical levels [13,16,17,18]. REM-locked responses in the visual hierarchy have led to the notion that REM sleep dreams may be accompanied by internally scanning a virtual sensorium: please see [19], and [16,20] for review and discussion of this ‘scanning hypothesis’, in relation to empirical evidence and dream reports. 

The implicit REM-locked generation of dream consciousness is supported by the finding that REM-locked activity is greater in the posterior left hemisphere [13], associating reports of dream experience with a ‘posterior hot zone’ activation [21]. The scanning hypothesis suggests that, REMs index the activity of the TPN, where the ‘task’ entails the generation and processing of a virtual reality in dreaming [22,23,24,25]. On this view, what we perceive as ‘reality’ when awake is based on the same virtual-reality models of the world—as in dreaming—but informed by sensory signals. This is consistent with concept of a ‘transparent phenomenal model’ of the world (with the self at its center), generated through multisensory integration, as an account of waking perception [23,26].

The focus of this paper is on the functional anatomy of processing in the extended visual hierarchy during REM sleep, in which scanning of a virtual sensorium is hypothesized to engage task-positive cortical areas—at sensory hierarchical levels—and disengage task-negative areas—at high levels. In brief, we show that certain cortical areas are exempt from widespread TPN activation time-locked to REMs and that these areas belong to DMN. 

Studying REM-locked brain responses can be contrasted with studying brain activation associated with REM sleep episodes [27,28]. Our event-related fMRI findings showed that REM-locked brain activity is distinct from baseline brain activity during REM sleep [16]. As REMs are contained in REM sleep episodes, brain activation, time-locked to REMs, look similar to brain activity patterns associated with REM sleep episodes but are attributed to events, as opposed to a brain state. This attribution can increase the statistical efficiency of detecting REM-related activity: a short six-minute study (with 43 REMs) was sufficient to evince a REM-locked activations and deactivations [13]. Furthermore, video-timing has clear advantages compared to conventional electrooculogram (EOG)-timing of REMs in fMRI studies [13]. For a time series analysis/event-related fMRI, sleep staging by EOG and electroencephalogram is unnecessary, but efficient timing of eye movements is crucial. REM sleep is an ideal state for studying the intrinsic functional architecture of the brain [13,16], because much of the external sensory input to the brain is attenuated in REM sleep [29], albeit incompletely (see [16] for review). In short, we anticipated that an event-related fMRI study of DMN-TPN interactions, time-locked to video-timed REMs, may further characterize the relationship between these intrinsic brain networks, in a way that is not confounded by exogenous stimuli. 

REM-locked brain responses fit comfortably with hierarchical predictive coding theory—a predominant theory of perceptual synthesis [16,18]. According to predictive processing view of perceptual inference, the brain infers—through hierarchical cortical processing—states of affairs in the world from the sensory signals that it samples [15,30,31]. Sampling through saccadic scanning is essential for active inference in wakefulness, i.e., generating, testing and serially updating ‘hypotheses’ about—or explanations for—the generation of sensory data [31]. Crucially, in enactive formulations of perceptual inference, “eye movements are both the cause and consequence of perception,” in both wakefulness and dreaming [18]. Direct comparison of REMs in wakefulness and in REM sleep show that both are associated with activations in the oculomotor and visuospatial attentional system [32]. In short, hierarchical (active) inference in the brain—time-locked either to waking saccades or to REMs in sleep—shares many formal similarities, in terms of neuronal message passing and hierarchical processing. Crucially, this formulation suggests that during active inference (i.e., the deployment of REM), attention increases message passing at lower cortical levels and attenuates processing at higher levels [33,34,35,36,37,38,39,40]. In summary, our previous findings of REM-locked TPN activation and recent evidence for TPN-DMN anticorrelation in REM sleep led us to predict DMN deactivations specifically time-locked to REMs—in accord with predictive processing accounts of perceptual synthesis.

## 2. Materials and Methods

### 2.1. Participants

Eleven healthy participants (5 females, mean age 24, range 19–37, 9 right-handed) (Table 1) gave written informed consent and joined this study, which was approved by the Johns Hopkins Medicine IRB. Their sleep was not deprived before this study. If a subject woke during scanning and could not fall asleep again, he/she was removed from the scanner. One out of fourteen participants could not fall asleep, and subsequently withdrew. A further two participants were not included in the analysis: one had unusually large and frequent, jerky head movements during scanning and we had technical difficulty timing REMs in the other. Video and sound monitoring was useful to determine if subjects were asleep: video monitoring evinced slow drifting eye movements, which is a sign of light sleep; breathing sound and snoring could also be monitored. Video monitoring of the whole body showed twitching of facial muscles, fingers and toes, which is a characteristic sign of REM sleep [13]. Each participant slept two consecutive nights in the MRI scanner from about 11 PM until they woke in the morning. In six participants, we were only able to record a sufficient number of REMs on the second night. Presumably, REM sleep deprivation in the first night built ‘REM pressure’ for the second. It is possible that head restraint—required for MRI studies—may suppress REM sleep. Each participant contributed 1–5 studies (24 in total). None of these previously published (24) studies [13] were excluded from our reanalysis and no new studies were added. 

### 2.2. Data Acquisition

To time REMs during sleep, we used a video camera with a zoom lens and an angled mirror mounted on the head coil. The cameras were aligned with the mirror to capture the subject’s eyes. We covered the part of the head coil that was in the camera’s field of view with black tape to ensure the bright white surface of the coil did not adversely affect the camera auto-gain adjustments (see the Supplementary Video S2).

The video camera setup is shown in Appendix A for 1.5 Tesla scanner studies and in Figure S1B for 3 Tesla scanner studies. For the 1.5 Tesla scanner studies, a Sony mini DV digital camcorder was placed on the shelf outside of the scanner room. The camcorder generated concurrent audiovisual recording of scanner noises and eye movements. This allowed the timing of REMs to be synchronized with the fMRI data. For the 3 Tesla scanner studies, the camera was placed inside the MRI scanner room, closer to the subject. The camera system was built by MRA, Inc. (Washington, PA, USA). The camera was mounted on an aluminum tripod and placed at the back of the of the scanner. During each scan, the PC received the triggers from a Philips fMRI trigger box through a serial port interface (also built by MRA, Inc.). The scanner was set to produce a trigger for every volume acquired. We ran E-Prime (Psychology Software Tools, Inc., http://www.pstnet.com, accessed on 12 February 2003–19 March 2004) on the PC with a protocol that counted the trigger pulses. On each trigger the PC emitted a short audio beep. Beeps indicate the timing of acquisition of the first of the 36 slices and repeat every two seconds (See TR was 2 s). Beeps were used to synchronize the timing of occurrence of rapid eye movements with the fMRI time series. Every 15 beeps, the paradigm spoke the count of pulses using a synthesized voice. This audio was recorded with the video of REMs; so at any point in the video one could determine the scan timing with unprecedented precision. 

We obtained fMRI data as soon as REMs were detected by video-monitoring, and for as long as they continued. MRI data were acquired at the F.M. Kirby Research Center for Functional Brain Imaging at Kennedy Krieger Institute. A 1.5 Tesla scanner was used for the earlier studies (May 2001–November 2002.) and a 3.0 Tesla scanner for the later studies (April 2003–March 2004.). The 1.5 Tesla data were acquired using a Philips Gyroscan scanner, using single-shot gradient-echo Echo-Planar Imaging (EPI). The fMRI imaging parameters were TR/TE/flip angle = 2000/35/90, 20 slices (parallel to the AC-PC line), slice thickness = 6 mm (no gap), matrix size = 64 × 64. The field of view was 240 mm, resulting in a nominal in-plane resolution of 3.75 × 3.75 mm. For all functional runs, the signal was allowed to reach a steady state over four initial volumes that were excluded from the analysis. The 3.0 Tesla data were collected on a Philips 3 T Gyroscan Intera scanner, using a multielement receiver coil to allow partial parallel imaging. SENSE EPI was used. Parameters were: TR/TE/flip angle = 2000/30/90; 36 slices (no gap); nominal resolution = 3.75 × 3.75 × 3.75 mm; SENSE factor = 2.0. Anatomical images were obtained using a T1-Fast Spin Echo sequence for one participant and Magnetization Prepared Rapid Gradient Echo (MP-RAGE) for all others. We combined 1.5 T studies and 3.0 T studies because the regional pattern of the REM-locked activation and deactivation were similar when using the two field strengths [13].

REMs were timed by visual inspection of video-recording by one of us (CC-HH). The DVD was played back on a PC to identify *jerky* eye movements, *either large or small,* under the closed eye lids, with mouse clicks. The ensuing timing of eye movements were recorded by the PC. This was repeated three times to ensure replication accuracy. REM timings that were within 0.5 s from other assignments were averaged and entered the analysis. If no eye movements were within the 0.5 s range, REM timing data were excluded from analysis.

Note that polysomnography (electroencephalograph, electromyograph and EOG)—the standard procedure to define REM sleep episode—is unnecessary for the above protocol, that identifies the neuronal correlates of REMs, not of REM sleep episodes. Identification of REMs alone, either by EOG or video monitoring, operation ensures REM sleep at the time of the occurrence of REMs.

### 2.3. Data Analysis

SPM2 was used for the original analysis and SPM8 for the reanalysis. Otherwise, original and reanalysis were identical. The fMRI timeseries were slice-time-corrected, realigned (after masking out the eyes), spatially normalized, and smoothed using a 6 × 6 × 6 mm full-width half-maximum Gaussian kernel. A high-pass frequency filter (128 sec) and a correction for temporal autocorrelation were applied to the time series. 

To study the neural correlates of REMs, we employed a rapid event-related Blood Oxygenation Level-Dependent (BOLD) fMRI design, where each REM constituted an event. Individual events (REMs) were modeled by a standard (gamma-variate-based) synthetic hemodynamic impulse response function. From the timing of REMs, we predicted the time course of fMRI signal (estimated hemodynamic changes following REMs), then identified the voxels that match this predicted time course using a general (convolution) linear model. We performed a standard ‘random-effects’ (summary statistics) analysis on the results of the 24 individual studies (using a one-sample t-test) to test for responses that are conserved over subjects.

To test the hypothesis that the TPN and DMN show reciprocal (and anticorrelated) behavior in relation to REM, the current reanalysis focused on areas that showed an attenuation of REM-locked activation—in relation to REM sensitive areas—and on regions that showed REM-locked signal decreases. 

## 3. Results

### 3.1. REM Characteristics

Table 1 shows the duration of sleep study analyzed (range 6.2–33.3 min) and number of REMs (range 43–417) for each study.

### 3.2. Attenuation of DMN Activity Associated with REMs

Figure 1A,B show that the regions associated with the DMN, i.e., PCu/PCC, RSC, medial prefrontal cortex, IPC, inferior temporal cortex, inferior frontal gyrus, and hippocampal formation, did not show REM-locked activation, *even at a low statistical threshold* (group analysis, *n* = 24, uncorrected *p* < 0.05). It should be noted that Figure 1 used a low threshold to show areas of attenuated REM-locked activation. Surface rendering of REM-locked activation (Figure 1C) shows that although REM-locked activation can be detected efficiently in TPN regions, they were not detected in cortical areas surrounded by the widespread cortical activation. 

### 3.3. REM-Locked fMRI Calculations in DMN Nodes

Analyses of the individual studies revealed a modest REM-locked fMRI signal decrease in DMN nodes: PCu/PCC in 12 out of 24 studies (Figure 2, Table 2) and RSC (notably, all in the left hemisphere) in 6 out of the 24 studies (Figure 3, Table 3). Strikingly, well circumscribed, small volumes (0.4–1.1 cm^3^) of RSC in the left hemisphere (RSC-Lt) were surrounded by activation (Figure 3, Table 3). In group analysis, REM-locked fMRI signal decreases were not seen in DMN nodes.

We investigated whether particular participant or study characteristics were associated with REM-locked signal decrease in PCu/PCC and in RSC-Lt (Appendix A). There were no significant effects for gender, participant age, REM sleep episode duration, REM count and REM density on REM-locked PCu/PCC signal decrease. Note that to study the neural correlate of REMs, we did not use REM count or REM density, but employed event-related fMRI analysis. At an individual subject level, we tested for the effects of several factors—including REM count or REM density—that might confound the REM-locked signal decrease in PCU/PCC (or RSC-Lt). We did not find a significant effect of REM count or REM density. On REM-locked RSC-Lt signal decrease, there was no significant effect except for gender (*p* < 0.05, Fisher exact test) and REM sleep duration (*M* = 16.1, *SD* = 6.6 for absent; *M* = 25.1, *SD* = 6.1 for present; *t* (22) = 2.9, *p* = 0.008). However, the effects of gender may have been due to one subject who contributed four of the six studies with REM-locked signal decrease in RSC-Lt. 

### 3.4. REM-Locked Periventricular Signal Decrease (PVSD)

The most consistent and robust REM-locked BOLD deactivation was found in periventricular areas (Figure 4, Table 4).

These results were observed with or without global normalization (using SPM5), which emphasizes the regionally specific dissociation of TPN and DMN responses. Global normalization is an issue because activations in one part of the brain can produce widespread deactivations elsewhere (if the global signal is subtracted from the image volume). The fact that regionally specific deactivations were seen with global normalization suggests these deactivations were not caused by global normalization. Results without global normalization are presented in this paper.

## 4. Discussion

### 4.1. Relatively Small Islands of Attenuated REM-Locked Cortical Activation

Our key finding is that in *group* analysis (n-24) the cortical areas exempt from widespread REM-locked activation (even at a lenient threshold) were restricted to the DMN, that is, inferior parietal cortex, PCu/PCC, RSC-Lt, hippocampal formation, inferior lateral temporal cortex and inferior frontal gyrus (Figure 1). Clearly, an absence of a response is more difficult to establish than presence of a response (either increase or decrease) using classical inference, and our findings need to be replicated. The *individual* analyses produced a complementary and consistent finding, showing a modest temporally-precise REM-locked decrease in key DMN nodes in a subset of independent studies (Figure 2 and Figure 3). This REM-locked fMRI signal decrease in a subset of DMN core nodes was replicated in a series of independent individual studies. In summary, the DMN regions identified by previous studies [1,2,3] feature reversal (Figure 2 and Figure 3) or attenuation of REM-locked cortical activation (Figure 1C). Our findings of REM-associated DMN deactivation and previous findings [6,12] suggest that functional organization of DMN and TPN is conserved over the awake state and dreaming.

*The characteristically-widespread brain activation time-locked to REMs* [13]—the backdrop of the relatively small areas of REM-locked deactivation—further supports the proposal that REMs index visual consciousness during REM sleep dreaming in a time-locked manner. Global availability of incoming sensory information to multiple brain systems may be essential for conscious experience [14,26]. We proposed that REMs are a prime example of active inference, which scans and generates visual percepts in dreaming [16,18]. Active inference underwrites conscious perception and requires global neural workspace [15] (p. 214). This suggests that DMN is spared from the REM-locked global cortical recruitment that indexes conscious experience. 

### 4.2. REMs Are Saccades

Our REM-PET [32] and REM-fMRI findings [13] suggest that the visual targets of REMs are largely chosen automatically (i.e., involuntarily), much like the targets of saccades during wakefulness. Both eye movements in wakefulness and in sleep engage the oculomotor circuit/visuospatial attentional system (i.e., TPN). Our REM-PET finding—that waking voluntary saccadic eye movements and REMs both activate oculomotor circuits—led us to the revised scanning hypothesis; namely, that REMs are visually guided saccades that reflexively explore dream imagery. In the REM-fMRI study, we expected to find extensive overlap between REM-locked activation and the systems that underwrite waking eye movements and visuospatial attention. We also expected to find REM-locked activation in the structures involved in visually-guided scanning during wakefulness. These expectations were confirmed. In short, we demonstrated, with PET [32], and functional MRI (fMRI) [13], that REMs are also saccades, which are associated with activation of the oculomotor circuit, in a time-locked manner. Scanning saccadic eye movements in wakefulness occur about four times a second, which interestingly is about the same frequency of bursts of PGO waves and REMs in sleep. 

### 4.3. Parallels with Hallucinogen-Induced Changes

Similarities between dreaming and hallucinogen-induced mental phenomenon and also between their neural correlates have been noted by several groups [41,42,43,44,45,46]. Our fMRI findings add new parallels to the list. Like REMs in sleep, hallucinogens also induce DMN deactivation [44,47] (see [42,48,49,50] for review), accompanied by the most robust activation in the primary visual cortex correlating with visual hallucinations [43,51] (see [50] for review). Additionally, hallucinogen-induced increased global integration [42,52,53] parallels REM-locked characteristically-widespread activation. Furthermore, psilocybin induces dose-dependent increase in frequency of saccades [54], interestingly, up to 4 per second, which is the same frequency of REMs. In summary, REM-locked DMN deactivation and hallucinogen-induced DMN deactivation lend support to each other—and the proposal that permissive deactivation of DMN leads to widespread cortical activation peaking in the primary visual cortex. 

### 4.4. Permissive Deactivation of DMN

The DMN has been considered as a high-level component of cortical hierarchies, providing top-down predictions and contextual guidance for lower TPN levels [41]. REM-locked multisensory integration and DMN deactivation is consistent with the proposal that DMN inhibits TPN under default conditions (i.e., under certain kinds of attentional set or perspective taking). Is REM-locked deactivation in DMN an epiphenomenon of TPN activation? It seems more plausible that DMN participates in the generation and scanning of a virtual reality by its permissive deactivation. REM-locked attenuation of DMN responses suggests that DMN is deactivated, while the brain generates perceptual narratives without external sensory input when dreaming (or while awake with external sensory input). This may extend the conceptualization of TPN vs. DMN as external vs. internal mode of attention, respectively [1]. Additionally, REM-locked DMN deactivation comports with hierarchical nature of predictive processing, and may suggest that DMN is a high-level cortico-cortical system that participates in model generation and updating by *permissive deactivation* of cortico-subcortical loops [41,42] as opposed to simply resting offline. In his review, which did not include studies of hallucinogens, Raichle noted that a particularly noteworthy facet of this hierarchical system: the DMN is situated at the top [55].

In terms of predictive processing—and in particular predictive coding—the anticorrelations between the TPN and DMN are usually interpreted in terms of distinct modes of inference [41,56,57,58,59]: in the context of an active sampling of the sensorium, lower levels of cortical hierarchies are rendered more excitable; thereby enabling neuronal message passing to higher hierarchical levels. This is sometimes cast in terms of affording prediction errors from lower levels greater precision, via synaptic gain control or excitation inhibition balance [15,39,60,61,62,63,64,65,66,67]. Conversely, in reflective, perspective-taking and introspective processing, greater precision is afforded higher levels in the DMN that renders the lower levels less influential in terms of belief updating at higher levels. Put another way, higher level DMN activity is relatively suppressed when lower levels are disinhibited to facilitate belief updating based upon sensory information (e.g., ascending prediction errors).

Permissive release of tonic inhibition underwrites oculomotor control (Appendix A; see [68] for review), suggesting that disinhibition may be a general neuronal mechanism in perceptual inference. Omnipause neurons in the brainstem, which tonically inhibit oculomotor neurons, cease firing about 16 ms. before saccade onset. The superior colliculus, which is a mesencephalic waystation situated between the pontine brainstem below it and the cortex above it, plays a pivotal role in saccade control. It releases this break and triggers saccades by inhibiting omnipause neurons. The superior colliculus itself is tonically inhibited by GABAergic input from the substantia nigra, pars reticulata; namely, the pallidum, the major output of the basal ganglia. Likewise, we propose that the DMN may tonically inhibit a network—generating eye movements and visual predictions—through parallel facilitatory, disinhibitory, and inhibitory basal ganglia circuits, i.e., cortico-striato-pallidal-subthalamo-nigral-brainstem/thalamo cortical pathways. Other potential routes for DMN’s tonic inhibition of the visual cortex are direct anatomical interconnections between DMN-visuoparietal cortices. Additionally, DMN and visual cortex are indirectly connected through DMN-superior colliculus-pulvinar-visual/parietal/premotor cortices and cortico-pulvinar-cortical thalamic loops including DMN-pulvinar-visuoparietal cortices. Studying the neural correlates of REMs may therefore be useful for testing the above hypothesis. 

It has been proposed that REM sleep dreaming may be associated with DMN activation [69,70,71], given that DMN is activated during self-referential cognition like mind wandering, daydreaming and simulation [69,70,71,72] and because the brain is active, but does not access environmental stimuli during REM sleep [73]. Meta-analysis of six PET studies of the neural correlates of REM sleep reported large, but incomplete, overlap with DMN [74]. As noted by authors, eight studies including three fMRI studies of REM-locked activation [13,75,76] were excluded in this meta-analysis. 

Our fMRI findings show that REM-locked brain activity is distinct from baseline brain activity of REM sleep [13,16]. Our study which employed time series analysis of video-timed REMs should afford greater statistical power than studies of neural correlates of REM sleep. Our findings indicate that REM-locked brain activities (excluding DMN) correlate with visual experience in dreaming. Our findings (and our proposal of permissive deactivation of DMN) argue against the proposal of DMN activation associated with REM sleep dreaming.

Additionally, our interpretation is inconsistent with the traditional task-negative view that DMN simply rests and is disengaged during task performance. However, recent findings lead to a task-positive view that DMN is actively involved in tasks [1,77,78], situated at the top of a hierarchy through interaction with sensorimotor network in lower hierarchy [56,78]. Furthermore, our interpretation comports with recent studies that suggested that DMN provides top-down predictions over a slower time scale [79,80,81] and receives contemporaneous phasic bottom-up signals [81]. 

### 4.5. Temporally-Precise REM-Locked Signal Decrease in DMN Nodes May Reflect Upward Inhibitory Messages

Permissive deactivation would require sustained attenuation of DMN activity, which would be reflected by a tonic decrease in activity, not temporally-precise deactivations. Signal decreases in PCC/PCu and RSC-Lt in the present study were temporally-precise in relation to REMs and may be related to inhibition of DMN, originating from structures with activation time-locked to REMs (i.e., message passing from low to high hierarchical levels), rather than permissive deactivation originating from DMN (i.e., message passing from high to low hierarchical levels) (Appendix A). Evidence supporting downward tonic [79,80,81] and upward phasic message passing to and from DMN [81] fit well with the present findings.

However, correlational studies (including ours) do not establish direction of causal influences. Employing magnetoencephalography (MEG) may support inferences about directed (effective) connectivity between TPN and DMN. MEG can measure neuronal responses in localized brain structures over a millisecond timescale around REM onset [82]. Additionally, it is yet to be established that DMN is at the top of the hierarchy.

REM-locked signal decrease in PCC/PCu and RSC-Lt was modest (uncorrected *p* < 0.05), but its validity is supported by relatively good localization of the signal decrease in those structures (Figure 2 and Figure 3) and by its replication in a series of independent studies—REM-locked signal decrease in PCU/PCC was replicated in twelve studies and that in RSC-Lt in six.

FIt is not clear, though, why this REM-locked signal decrease was found in only a subset of studies. Four out of six REM-locked deactivations in RSC-Lt were in the same participant, suggesting that a certain characteristic of this participant contributed to this effect. It would require a larger number of subjects to study what individual characteristics correlate with this kind of REM-locked response in RSC-Lt. Finally, it is not clear why REM-locked signal decreases was found in PCu/PCC and RSC-Lt, but not other DMN nodes. PCu/PCC and RSC may be the main hub of DMN starting from early infancy, both in emergent and established DMN regions [83].

### 4.6. DMN Plays a Role in Aberrant Generation of Perceptual Content

The functional organization of DMN and TPN is altered in mental disorders including schizophrenia, autism, ADHD, and Alzheimer’s disease: please see [81,84,85,86,87,88] for review. More specifically, it has been reported in healthy aging, traumatic brain injury, ADHD, autism and schizophrenia that PCC fails to deactivate as cognitive load increases [81]. 

We proposed that REMs furnish a unique probe into normal and abnormal neurodevelopment of consciousness [16]. Particularly, as REMs, which index generation of visual consciousness [13,16,18,89], appear to be associated with deactivation of DMN. Studying changes time-locked to REMs affords statistical efficiency (determined by the number of data points in the timeseries). Finally, REMs are task-free probes. Thus, REMs can be employed in cognitively-impaired patients and infants (and even in animals) who are unable to cooperate with conventional brain activation paradigms [16].

### 4.7. REM-Locked PVSD Indicates REM-Locked Cerebral Vasoconstriction

Periventricular signal decrease (PVSD) occurred in all 24 studies [13]. Group analysis confirmed a robust REM-locked signal decrease around the walls of the lateral ventricle, the third ventricle, and the fourth ventricle [13] (Figure 4, Table 4). REM-locked PVSD is not an artifact, but a veridical BOLD change [13]. Interestingly, anticorrelation of BOLD fMRI signals between periventricular areas (around large cerebral veins) and visual cortex was found in waking resting state (using a 7T scanner), suggesting that PVSD does not reflect neuronal deactivation [90]. Inhalation of CO_2_, potent vasodilator, induced PVSD, probably because vasodilation replaced high-intensity signal CSF with lower intensity signal blood [91]. Regional patterns of CO_2_-induced PVSD are similar to that of REM-locked PVSD. Additionally, hypocapnia created by deep breathing causes global BOLD signal decrease, and anticorrelated, but positive, periventricular signal increase, further suggesting that PVSD is mainly driven by relative CSF/blood volume changes [92]. Recent fMRI studies in wakefulness [90,91,92] revealed that REM-locked PVSD indicates REM-locked cerebral vasodilation. To our knowledge, REM-locked cerebral vasodilation has not been previously reported. 

Anticorrelation of cortical activity and CSF flow (as measured by fMRI) during non-REM sleep has recently been reported [93]. It is possible that REM-locked neuronal activation leads to increases in cerebral blood flow and volume, with a concomitant reduction in CSF flow (and fMRI signal in CSF). However, the peak of REM-locked PVSD is clearly localized in the ventricular wall, indicating that REM-locked PVSD is mainly due to vasodilation (and consequential partial volume effects) rather than reduction in CSF flow, because the flow of fluid will be faster in the center of the tube than at the tubal wall. Furthermore, the time scale of CSF inflow is different: whereas pulsatile inflow of CSF—that is tightly temporally coupled to reduction in cortical activity—occurs approximately every 20 s, PVSD is time-locked to temporally-precise REMs in sleep, which occur on a faster time scale.

### 4.8. Vascular vs. Neuronal Change in Other Structures

Similarly, REM-locked signal decrease in white matter (Figure 4A) appears to be of vascular origin. It has been reported that hypercapnia induced a reduction in blood flow to and a reduction in fMRI BOLD signal in periventricular and frontal white matter [94], where REM-locked signal decrease was found. It has been suggested that it was a white matter steal phenomenon occurring at the arterial border zones [94]. The failure to detect of REM-locked signal increase in the homologous parts of both occipital lobes (Figure 1C) may reflect a lack of neuronal responses. For example, in nonhuman primates fMRI BOLD deactivations in visual cortex have been found several millimeters away from the positively responding primary visual cortex (V1) [95]. 

REM-locked deactivation in RSC-Lt (Figure 1A and Figure 3, Table 3) should be interpreted with caution. It was found in only a quarter of the studies. These deactivations may be of vascular origin, as they were found in CSF-rich region, where vasodilation replacing high-intensity CSF causes signal decrease [90,91,92]. It is unlikely, however, that REM-locked vasodilation occurs only at the left RSC-CSF border. If it indeed reflects neuronal deactivation, it may answer the question whether RSC is a component of DMN or TPN. Most lesion studies suggest that the RSC in the right hemisphere (RSC-Rt) is involved in topographic orientation or spatial navigation (most brain imaging studies show bilateral activation [96], probably because RSC is relatively small and the right and the left RSCs are juxtaposed) [13]. Waking resting state fMRI has shown that RSC is the part of DMN [97]. Our finding may suggest that RSC-Rt belongs to TPN and RSC-Lt belongs to DMN.

### 4.9. Timing REMs by Visual Inspection of Video-Recording

In a study of the neural substrates of REMs, timing of REMs is as crucial as fMRI measurements. Electrooculogram (EOG) is conventionally used to detect REMs, but rapidly changing magnetic fields during MRI confound the EOG signal. Thus, MRI scanner artifacts need to be removed by a filter [75,76] and filtering reduces the number of eye movements detected, especially small amplitude eye movements. It causes disadvantage as small and large eye movements may have almost the same effect on the fMRI signal [98]. Video monitoring detects even small eye movements under closed eye lids (see supplementary videos). Video monitoring detected approximately four times as many REMs compared to EOG in fMRI studies [13]. EOG measures electric currents induced by movements of the electrically charged eyeball, whereas video-monitoring directly measures eye movements. Eye movements detected by video monitoring and by EOG are fundamenatlly the same. Indeed, we showed that REM timings derived from EOG and video recording were in excellent agreement [13] (Figure 5). Video-timing of REMs and time series analysis of REMs should afford uniquely strong statistical power. 

The accuracy of timing REMs by visual inspection of video-recording—compared with EOG timing—is sufficient for event-related fMRI study (the events being REMs) (Figure 5). However, objective classification of REMs is not possible with human visual inspection. One could consider automated (machine learning) classification. For example, a feature-tracking algorithm was successfully employed for quantitative analysis of videos of closed eyes of sleeping lizards [99]. 

### 4.10. Superb Localization Capacity of fMRI of Video-Timed REMs (vtREM-fMRI)

The group analysis showed undetectable REM-locked activation in RSC-Lt that contrasted dramatically with adjacent robust REM-locked activation on the right [13]. Analyses of individual studies revealed REM-locked deactivation in well-circumscribed, small volumes of RSC-Lt, surrounded by widespread activation in some studies (Figure 3). This finding attests to the capacity of vtREM-fMRI to localize REM-locked BOLD signal increase [13] or decrease. vtREM-fMRI provided clear localization of the peak REM-locked signal change in thin structures like thalamic reticular nucleus, claustrum and periventricular structure—and detected small circumscribed deactivation surrounded by diffuse activation. These effects have not been observed in the fMRI studies that timed REMs with electrooculograms (EOG)—the traditional method of detecting REMs in fMRI studies [75,76]. The reason for this localization capacity may be that video monitoring detects approximately four times as many REMs compared to EOG in fMRI studies [13].

### 4.11. Attenuation of REM-Locked Activation in the Hippocampal Formation May Explain Forgetting of Dreaming and Waking Experience

This (Figure 1B) is an interesting finding because it is relatively easy to activate hippocampus with paradigms that engage encoding or retrieval [100]. Only mental activities that engage mentation but are not mnemonic—such as rapid perceptual or cognitive judgments—are associated with substantially lower hippocampal responses [100]. 

Our findings suggest that unlike in wakefulness, REM-locked perceptual experience [13] is not encoded, i.e., the dreamer is actively engaged in interacting with the virtual environment [13,62], but the brain is not encoding it. This notion is supported by reports that informational flow from the hippocampus to neocortical areas is blocked during REM sleep [101,102]. Dreams are often as vivid and real as in wakefulness but are quickly forgotten. It has been proposed that dreams are forgotten because of repression of the retrieval of stored memory [103] (disputed in [104]) or due to impoverished access to stored memories [105]. Our findings suggest that dreams are quickly forgotten because dream experiences are not encoded at their inception [106]. 

Our REM-associated hippocampal finding may have implications for forgetting of waking experience as well. Hippocampal theta waves—concurrent with exploratory activity in wakefulness—also occur in REM sleep [107]. It has been proposed that sensorimotor experience initially encoded in wakefulness is later consolidated offline in sleep [101,107]. GABAergic neurons involved in hippocampal theta rhythm generation have a role in memory consolidation in REM sleep [108]. Conversely, a group of neurons in the hypothalamus which are most active in REM sleep inhibit hippocampal neurons and are involved in active forgetting [109]. Both memory consolidation and active forgetting may occur during REM sleep [109]. 

### 4.12. Limitations of the Present Study

The detection of fMRI deactivation in the DMN in most individuals (or studies)—using uncorrected statistical thresholds—in the absence of statistically significant decreases at the group level, does not provide compelling evidence that the DMN is systematically deactivated in association with REMs in sleep. As such our conclusions should be taken with caution. In other words, although REM-related deactivation of the DMN, relative to fluctuations in fMRI signal, is evident in some subjects, this effect is small in relation to its between-subject (or study) variability.

Another limitation is that we did not account for the effects of scanning different subjects at different field strengths. Although one can demonstrate consistency across 1.5 Tesla and 3 Tesla studies—and across participants—(see Figures 5 and 8 in Hong et al. 2009 [13]), it is possible that heteroscedasticity (i.e., differences in error variance) may have compromised the statistical efficiency of our analyses. As these previously unreported findings have significant implications, our group and individual results both call for replication. 

## 5. Conclusions

Although we present new data analyses in this paper, this is primarily a hypothesis paper, which uses suggestive empirical results as a vehicle to substantiate a particular theoretical treatment. Although one cannot infer the absence of activation, using classical inference to accept the null hypothesis of no REM-locked responses in certain brain regions, the convergent findings are taken as evidence for attenuation of REM-locked activation in the DMN. Our findings suggest that the functional integration of the DMN and TPN is essentially the same during the awake state and the REM dream state. REM-associated attenuation of DMN responses is consistent with permissive deactivation of DMN (i.e., backward message passing from higher to lower cortical and subcortical hierarchical levels that may instantiate a form of disinhibition)—during sensory engagement with a real or virtual sensorium—that holds during both sleep and wakefulness. The DMN may be involved in the generation of REMs and visual imagery via tonic permissive disinhibition. Temporally-precise REM-locked signal decrease in precuneus and adjacent retrosplenial cortex (RSC-Lt) may be caused by forward message passing from lower to higher hierarchical levels. Bidirectional message passing is a requisite for hierarchical processing and there are canonical routes established for it. 

REM-locked deactivation findings further support the connection of REM-locked activation findings to hierarchical predictive coding [16]: REM-locked activation findings endorse the idea that REMs underwrite active inference, are consistent with REM-locked cholinergic activation, and speak to an intimate relationship between perception, action and attention—in both waking and sleep.

Studying the neural correlates of video-timed REMs led to some novel findings. These findings call for replication, particularly with the millisecond temporal resolution of MEG. However, suggestive evidence for REM-associated DMN deactivation strengthens our claim that REMs furnish a potentially fruitful and unique probe into normal and abnormal consciousness.

In summary, REMs may offer a promising probe to study conscious processing, especially in visual perception [16]. Practically, event-related fMRI studies employing REMs have high statistical efficiency. This paradigm should facilitate analysis of within-subject analyses, including correlations of REM-locked responses with fluctuation in conscious level and dream reports collected via a serial awakening [110] as well as longitudinal studies of typical development of visual perception in animals and infants—as REMs are natural, task-free probes—and its abnormal development in persons with autism and schizophrenia. 

## Figures and Tables

**Figure 1 brainsci-11-01528-f001:**
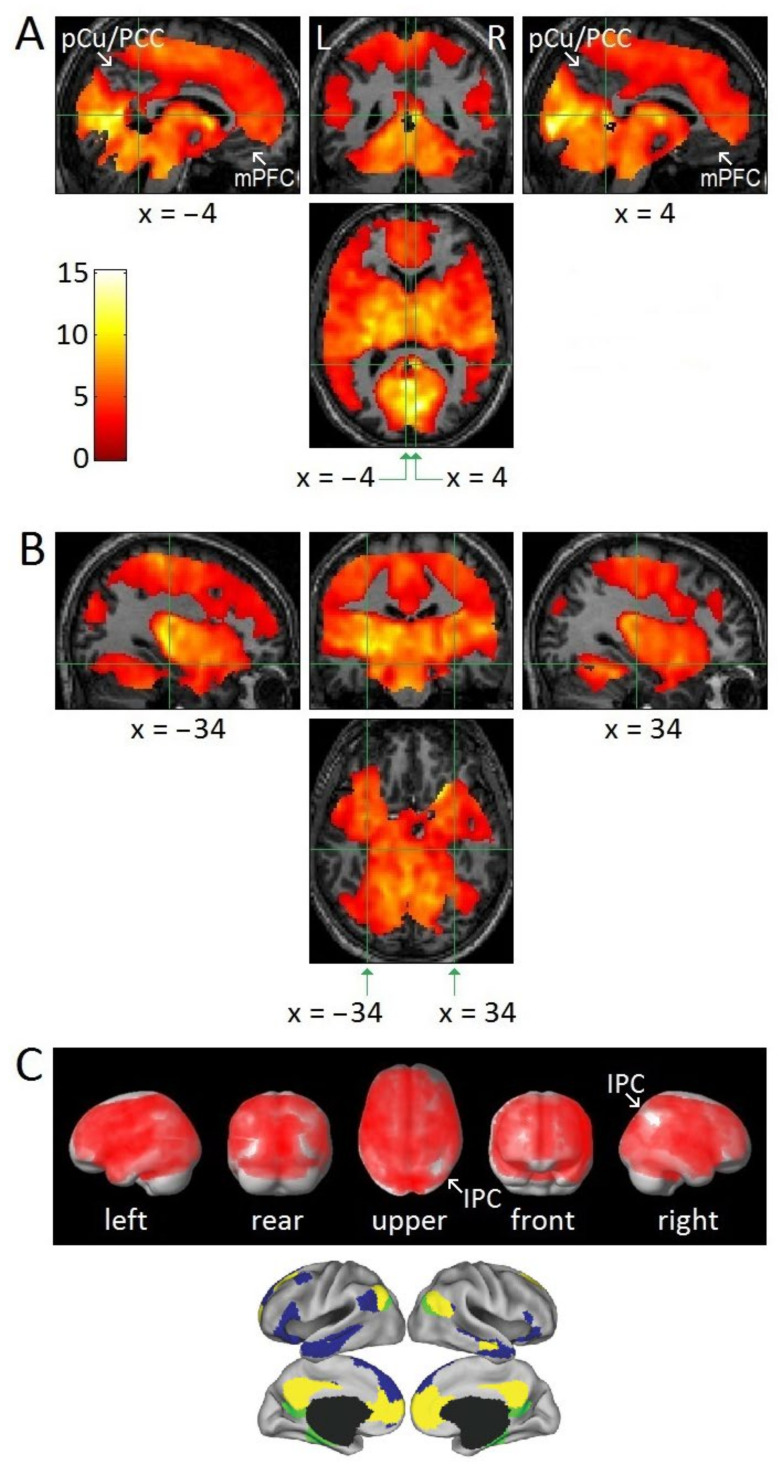
Areas of undetectable or attenuated REM-locked activation. Uncorrected level of *p* < 0.05 was used to show areas of undetectable or attenuated activation, even at this low threshold: random-effects analysis (group analysis, n = 24, one-sample *t*-test). Green lines show the locations of the other orthogonal views. In (**A**,**B**), the upper left and right panels show sagittal views, the upper middle panel, a coronal view, and the lower panel, a axial view; the statistical parametric maps are superimposed on anatomical images and show an absence of significant REM-locked effects in several regions: (**A**) PCu/PCC and medial prefrontal cortex (mPFC) on the left and the right hemisphere. Crosshair at the RSC-Rt (4,−46,12) and at the corresponding location in the contralateral hemisphere (−4,−46,12). The pocket of no activation in RSC-Lt contrasts dramatically with the adjacent robust activation in RSC-Rt (*t* = 10.5). Coronal and axial views prepared with SPM2 were previously published [13]. (**B**) Hippocampal formation on the left and the right hemispheres. Crosshairs at (−34,−24,−12) and (34,−24,−12). Note the absence of activation in much of the hippocampal formation. (**C**) REM-locked activation effects projected on to surface rendering of a template brain. Note further the absence of activation in IPC on the right hemisphere and in the homologous areas of the occipital lobes on both sides at this threshold: attenuated activation in IPC on the left hemisphere and bilaterally in inferior lateral temporal cortex and inferior frontal gyrus. Bottom figures were adapted from the large-scale meta-analyses of Andrews-Hanna et al. (2014) with the permission. DMN core (yellow), dorsal medial subsystem (blue), medial temporal subsystem (green).

**Figure 2 brainsci-11-01528-f002:**
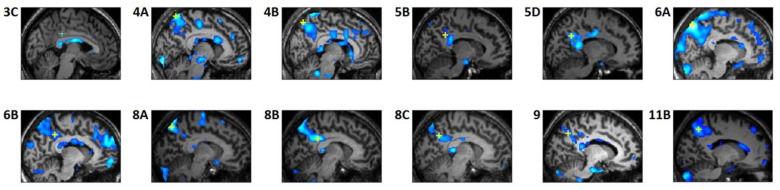
REM-locked fMRI signal decrease in precuneus/posterior cingulate cortex. Each study constitutes an independent within-subject study. Labels indicate participant and study number, e.g., participant 5 contributed five studies (5A–5E), 5B is the second study and 5D is the fourth study. Talairach coordinates and *t* values of the maximum signal decrease voxels (yellow crosshair) are shown in Table 2. All panels show sagittal views. All studies were thresholded at uncorrected *p* < 0.05 (see color scales in Figure 3) and additionally at a spatial extent of <5 contiguous voxels.

**Figure 3 brainsci-11-01528-f003:**
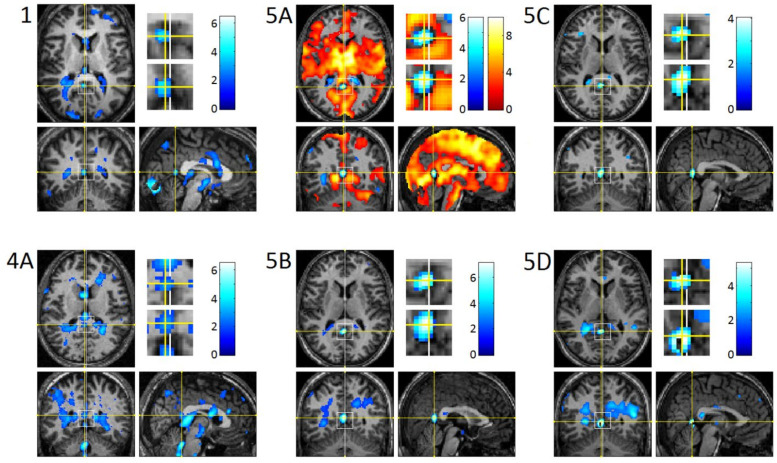
REM-locked fMRI signal decrease in RSC-Lt. The labels are the same as those in Figure 2. Each study constitutes an independent within-subject study. Yellow cross hairs are on the voxels of maximum signal decrease in RSC. Talairach coordinates and *t* values of the maximum deactivation voxels are shown in Table 3. Lines show location of the other views. Squares in the axial and coronal views are magnified. White vertical lines in the centers of the magnified views indicate the mid-sagittal plane. The squares in 5A–D were placed in the same location. All studies were thresholded at uncorrected *p* < 0.05 and additionally at a spatial extent of <5 contiguous voxels. The clusters were well circumscribed, 0.4~1.1 cm^3^. In all six studies, the clusters centered on the RSC-Lt. REM-locked activation maps are overlaid in 5A to show that RSC deactivation is surrounded by activation. Two further studies (4B and 7) also showed circumscribed REM-locked signal decrease near RSC-Lt, but this was not included because the majority of these fell on cerebrospinal fluid (CSF) areas.

**Figure 4 brainsci-11-01528-f004:**
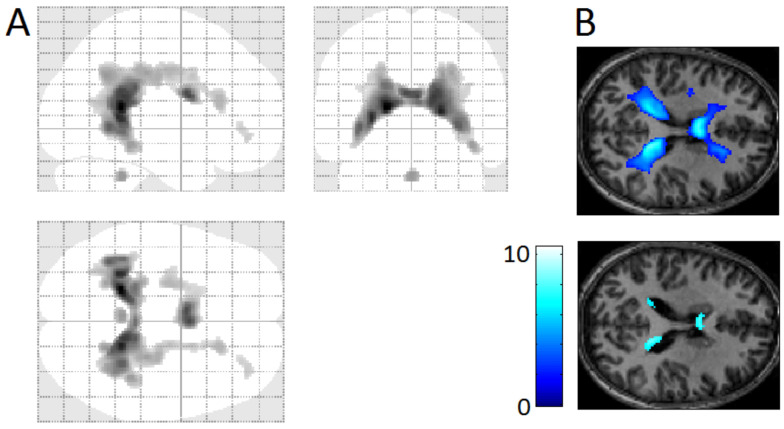
REM-locked periventricular BOLD signal decrease (group analysis, n = 24). Significance was thresholded at *p* < 0.05, uncorrected (T = 1.7) and additionally at a spatial extent of >5 contiguous voxels. (**A**) orthogonally oriented ‘glass brain’ views. (**B**) the lower panel was thresholded higher at *p* < 0.05 corrected for multiple comparison (T = 6.0) to show that the robust signal decrease centers around walls of the lateral ventricle. Results of analyses with SPM2 were previously published [13].

**Figure 5 brainsci-11-01528-f005:**
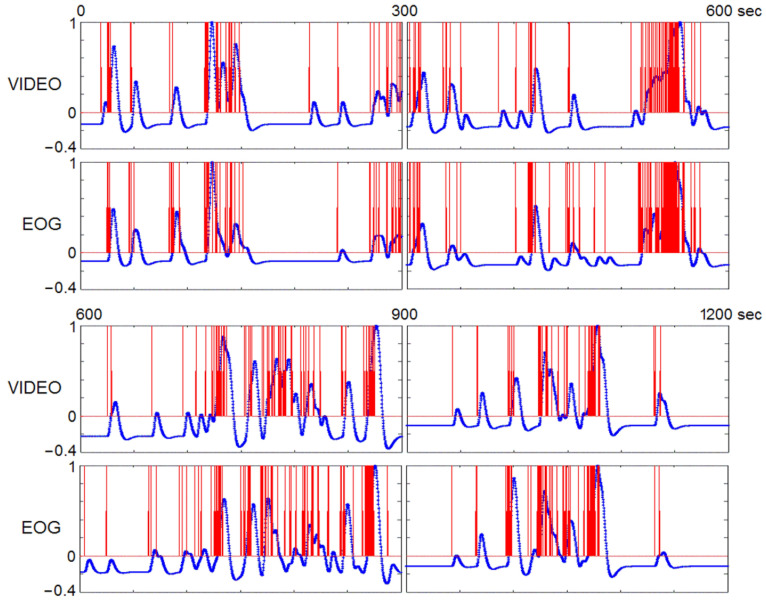
Timing of REMs using videorecording versus EOG. Eye movements were timed from videorecording and EOG that were obtained simultaneously in an out-of-magnet study (shown in red), then convolved with hemodynamic function (shown in blue). The hemodynamic impulse response had a six-second latency. There are five data points in one second. Regardless of the method of timing eye movements (videorecording, EOG, or other), the step right before correlation with fMRI data is to convolve with hemodynamic function. Therefore, these common end products (the curve derived from videorecording and the curve from EOG-recording, both shown in blue) were used in comparing video and EOG methods. The correlation between those two curves was highly significant (*r*_s_ = 0.84; *p* < 0.00005). Adapted from Hong et al. (2009) with the permission from Hong.

**Table 1 brainsci-11-01528-t001:** Participant and REM characteristics.

Participant	Study	Age	Sex	Analyzed Sleep Duration ^a^	REM Count ^b^	Inter-Scan Interval ^c^
1		25	M	20.0	230	
2		19	F	27.3	389	
3	a	18	F	14.6	174	
	b			12.1	108	5 months
	*c*			*6.2*	*43*	17 months
	*d*			*13.6*	*189*	19 months
4	a	37	M	22.1	105	
	b			10.8	103	20.9 h
5	a	20	M	33.3	192	
	b			29.1	156	2.1 h
	c			17.7	283	25.5 h
	d			28.4	324	27.7 h
	*e*			*20.5*	*334*	6 months
6	a	22	F	12.0	76	
	b			11.4	122	23.4 h
	c			23.2	309	26.1 h
*7*		24	F	*16.6*	*236*	
*8*	*a*	22	M	*22.7*	*154*	
	*b*			*25.7*	*337*	23.4 h
	*c*			*7.8*	*70*	25.3 h
*9*		23	M	*26.2*	*417*	
*10*		23	F	*9.0*	*76*	
*11*	*a*	25	M	*13.2*	*170*	
	*b*			*16.4*	*223*	2.1 h

Some participants had more than one scan (scans labelled chronologically). The 3 Tesla scans are shown in italic. ^a^ Duration of sleep analyzed (in minutes). ^b^ Number of rapid eye movements. ^c^ Time elapsed after the start of the first scan. Adapted from Hong et al. (2009) with the permission from Hong.

**Table 2 brainsci-11-01528-t002:** REM-locked fMRI signal decrease in precuneus/posterior cingulate cortex.

Study	Maximum Signal Decrease Voxel			
	x	y	z	*t*
3C	4	−31	35	2.2
4A	8	−61	68	4.7
4B	−6	−67	57	4.5
5B	12	−47	34	2.1
5D	14	−55	32	3.2
6A	−10	−71	53	7.1
6B	10	−45	39	3.6
8A	10	−69	53	4.3
8B	8	−41	32	5.6
8C	8	−58	38	3.8
9	12	−60	42	2.9
11B	−14	−60	49	5.2

This table shows Talairach coordinates and *t* values of the maximum signal decrease voxels. For all *p* < 0.05, uncorrected for multiple comparisons. Cluster sizes were not provided as none of the clusters were as well circumscribed as RSC-Lt clusters.

**Table 3 brainsci-11-01528-t003:** REM-locked fMRI signal decrease in RSC-Lt.

Study	Maximum Signal Decrease Voxel				Cluster Size
	x	y	z	*t*	
1	−2	−52	14	3.8	55
4A	−2	−42	17	3.1	9257
5A	−2	−54	14	6.0	86
5B	0	−52	14	7.1	139
5C	−2	−52	14	4.0	95
5D	−2	−52	8	5.3	107

This table shows Talairach coordinates and *t* values of the maximum signal decrease voxels. For all *p* = 0.05, uncorrected for multiple comparisons. Cluster size shows number of voxels in the cluster (voxel size = 2 mm × 2 mm × 2 mm). Clusters were well circumscribed, but the RSC cluster was connected to the large cluster in 4A.

**Table 4 brainsci-11-01528-t004:** REM-locked periventricular BOLD signal decrease (group analysis, n = 24).

Study	Maximum Signal Decrease Voxel				Cluster Size
	x	y	z	t	
LV	−20	−42	13	10.5	11,468
3V	0	−6	−8	2.5	12
4V	−2	−44	−25	5.6	91

This table shows Talairach coordinates and *t* values of maximum PVSD. For all *p* = 0.05, uncorrected for multiple comparisons. Cluster size shows number of voxels in the cluster (voxel size = 2 mm × 2 mm × 2 mm). LV: lateral ventricle; 3V: 3rd ventricle; 4V: 4th ventricle.

## Data Availability

The raw data supporting reported results will be made available by the corresponding author, without reservation.

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
