# Peer review of "fMRI Evidence for Default Mode Network Deactivation Associated with Rapid Eye Movements in Sleep"

_brainsci, 2021, doi:10.3390/brainsci11111528_

Round 1
Reviewer 1 Report
This manuscript reports an event-related reanalysis of fMRI data, previously collected during REM sleep, that is aimed at assessing whether the default mode network (DMC) shows brief deactivations during individual REM events. The authors report results of a one-sample t-test that show statistically significant, positive REM-related activation effects throughout almost the whole brain except for a few DMN regions, as well as separate whole-brain tests from a subset of individual within-subject studies that show deactivation patterns in some DMN regions that are qualitatively similar to one another. They claim that these results provide evidence for DMN deactivation during REM events. Despite the uniqueness of the data set, the authors’ careful and painstaking efforts to perform an event-related analysis on temporally precise REM events, and the interesting topic, I have serious concerns about the analyses the way they are interpreted.
First and foremost, I am puzzled as to why the primary statistical test that could provide evidence for the hypothesis of DMN deactivation was apparently not conducted or reported: a group-level one-sample t-test for negative (rather than positive) activation in response to REM events. In the absence of such a test, the two main findings that are reported instead do not provide compelling evidence for DMN deactivation. As the authors note, failing to reject the null hypothesis of DMN activation in the reported one-sample t-test for positive activation does not actually provide statistical evidence for the null, let alone for deactivation. The other result highlighted, qualitative similarities between deactivation patterns in a handful of within-subject studies, is also not compelling because individual-level activation maps are well known to be very noisy (especially at the extremely liberal, and uncorrected, statistical threshold that is used in this study), and searching them for qualitative similarities is therefore bound to reveal patterns in a handful of subjects that can be argued to support almost any hypothesis. Without a group-level statistical test of deactivation, it is impossible to know whether these patterns occur with statistical regularity in a subset of individuals, or simply reflect confirmation bias. In my view, the results that are currently reported are, at best, “not inconsistent with” the possibility of DMN deactivation during REMs.
Another concern is whether this data set is even appropriate for standard group-level analyses. Since the within-subject studies are not independently sampled (they are nested within subjects and also within scanner type/field-strength) they likely violate the assumptions of a [traditional random effects] group-level t-test. Some kind of multi-level model that accounts for nesting (e.g., random intercepts for site and scanner type) may be more appropriate.
Finally, the connection made between DMN deactivation and hierarchical predictive coding theory in the Discussion seems somewhat abstract. DMN anticorrelation with task-positive networks has been repeatedly observed in many contexts and could potentially be explained by an array of other theories. Why would this observation lend particular support to hierarchical predictive coding theory? Furthermore, what set of observations in this data set would, if observed, be inconsistent with, or falsify, hierarchical predictive coding theory?
I also have a few more minor concerns:
- “one out of 13 participants withdrew” but results are reported for only 11; what happened to the 12th?
- Were there any motion-related exclusions, either at the level of within-subject studies or the scrubbing of individual frames?
- Many DMN regions are midline regions, but Figure 1C only shows the outer surfaces of the brain and not the midline.
- Maybe I am interpreting Figure 1 incorrectly, but it seems to me like medial frontal cortex is very clearly activated, which is inconsistent with what is reported in the text.
Reviewer 2 Report
The study by Hong et al. investigate the association between REM and multisensory integration of signals in the brain. The present manuscript builds up on previous studies published by the authors, and basically performs a more thorough analysis of previously captured MRI data in healthy subjects. This is an interesting and well designed study, and the authors are clear on their predictions and pitfalls of experiments. Overall, this reviewer does not have major concerns that need to be addressed, as, in my opinion, the authors acknowledge all limitations of the data collection and interpret findings with caution.
The only comment this reviewer has regards the use of direct quotes from previously published manuscripts, which I regard as inappropriate for the scientific publication format, even when citations are provided. My advice is to remove such direct quotes and simply paraphrase the findings, maintaining the citation, for a more appropriate and clean presentation of findings / discussion.
Author Response
Reviewer 2
The study by Hong et al. investigate the association between REM and multisensory integration of signals in the brain. The present manuscript builds up on previous studies published by the authors, and basically performs a more thorough analysis of previously captured MRI data in healthy subjects. This is an interesting and well designed study, and the authors are clear on their predictions and pitfalls of experiments. Overall, this reviewer does not have major concerns that need to be addressed, as, in my opinion, the authors acknowledge all limitations of the data collection and interpret findings with caution.
The only comment this reviewer has regards the use of direct quotes from previously published manuscripts, which I regard as inappropriate for the scientific publication format, even when citations are provided. My advice is to remove such direct quotes and simply paraphrase the findings, maintaining the citation, for a more appropriate and clean presentation of findings / discussion.
We thank the reviewer for acknowledging the strengths in the manuscript.
We agree with the reviewer 2 regarding direct quotes from citations. We have tried to paraphrase all of the direct quotes, but for one, short, clear direct quote, we could not find better replacement.
Round 2
Reviewer 1 Report
This is a revision of a manuscript I previously reviewed that reports event-related analyses of DMN and other cortical activity during REMs. I appreciate the authors' responses to my initial concerns, including now reporting the null results from the analysis that is the primary test of their hypothesis (the group-level test of whether DMN regions show significant deactivation in response to REMs), adding details about how the reported results relate to the hierarchical predictive coding theory, and clearing up ambiguities about participant exclusions and the effects shown in figures. I still strongly disagree with the authors’ claim that the lack of significant positive activations in group-level analyses and the presence of apparent deactivations in a subset of individual-level tests with uncorrected and liberal statistical thresholds suggests, in any compelling way, that the DMN is deactivated during REMs. Further, I do not find their argument for ignoring the nesting of the data by scanner and by participant in their analyses to be compelling. Just because localization of certain effects is found to be “clean” when this nesting is ignored does not mean that systematic differences in activation magnitude do not exist across scanners and across participants or that such differences are not potentially biasing significance test statistics in a meaningful way. However, I recognize that ultimately it is the responsibility of the readers of this report to judge the evidence for these claims for themselves. Therefore, I will defer to the decision of the editor and let readers independently evaluate the report’s claims if it is published, although I still think that any effort the authors can make to further qualify, and describe potential weaknesses of, these claims would improve the paper.
Author Response
We appreciate these suggestions that help to improve our paper by providing a more conservative, circumspect summary of our conclusions.
We added a new section to make the limitation of our present study clear.
4.12. limitations of the present study
The detection of fMRI deactivation in the DMN in most individuals (or studies)—using uncorrected statistical thresholds—in the absence of statistically significant decreases at the group level, does not provide compelling evidence that the DMN is systematically deactivated in association with REMs in sleep. As such our conclusions should be taken with caution. In other words, although REM-related deactivation of the DMN, relative to fluctuations in fMRI signal, is evident in some subjects, this effect is small in relation to its between-subject (or study) variability.
Another limitation is that we did not account for the effects of scanning different subjects at different field strengths. Although one can demonstrate consistency across 1.5 Tesla and 3 Tesla studies—and across participants—(see Figures 5 and 8 in Hong et al. 2009), it is possible that heteroscedasticity (i.e., differences in error variance) may have compromised the statistical efficiency of our analyses. As these previously unreported findings have significant implications, our group and individual results both call for replication.
